

**Parametrization consequences of constraining soil organic**
**matter models by total carbon and radiocarbon using long-**
**term field data**
**L. Menichetti[1], T. Kätterer[2] and J. Leifeld[1]**
[1]{ Agroscope, Climate / Air Pollution Group, Reckenholzstrasse 191, CH 8046 Zürich,
Switzerland}
[2]{Swedish University of Agricultural Sciences (SLU), Department of Ecology, Box 7044,
75007 Uppsala, Sweden}
Correspondence to: L. Menichetti (Lorenzo.Menichetti@agroscope.admin.ch)
**Abstract**
Soil organic carbon (SOC) dynamics result from different interacting processes and controls
on spatial scales from sub-aggregate to pedon to the whole ecosystem. These complex
dynamics are translated into models as abundant degrees of freedom. This high number of not
directly measurable variables and, on the other hand, very limited data at disposal result in
equifinality and parameter uncertainty.
Carbon radioisotope measurements are a proxy for SOC age both at annual to decadal (bomb
peak based) and centennial to millennial time scales (radio decay based), and thus can be used
in addition to total organic C for constraining SOC models. By considering this additional
information, uncertainties in model structure and parameters may be reduced.
To test this hypothesis we studied SOC dynamics and their defining kinetic parameters in the
ZOFE experiment, a >60-years old controlled cropland experiment in Switzerland, by
utilising SOC and $SO^{14}C$ time-series. To represent different processes we applied five model
structures, all stemming from a simple mother model (ICBM): I) two decomposing pools, II)
an inert pool added, III) three decomposing pools, IV) two decomposing pools with a
substrate control feedback on decomposition, V) as IV but with also an inert pool. These
structures were extended to explicitly represent total SOC and $^{14}C$ pools.





The use of different model structures allowed us to explore model structural uncertainty and
the impact of $^{14}$C on kinetic parameters. We considered parameter uncertainty by calibrating
in a formal Bayesian framework.
By varying the relative importance of total SOC and SO$^{14}$C data in the calibration, we could
quantify the effect of the information from these two data streams on estimated model
parameters. The weighing of the two data streams was crucial for determining model
outcomes, and we suggest including it in future modelling efforts whenever SO$^{14}$C data are
available.
The measurements and all model structures indicated a dramatic decline in SOC in the ZOFE
experiment after an initial land use change in 1949 from grass- to cropland, followed by a
constant but smaller decline. According to all structures, the three treatments (control, mineral
fertilizer, farmyard manure) we considered were still far from equilibrium. The estimates of
mean residence time (MRT) of the C pools defined by our models were sensitive to the
consideration of the SO$^{14}$C data stream. Model structure had a smaller effect on estimated
MRT, which ranged between 5.91 and 4.22 years and 78.93 and 98.85 years for young and
old pool, respectively, for structures without substrate interactions.
The simplest model structure performed the best according to information criteria, validating
the idea that we still lack data for mechanistic SOC models. Although we could not exclude
any of the considered processes possibly involved in SOC decomposition, it was not possible
to discriminate their relative importance.

## 1   Introduction

The dynamics of soil organic carbon (SOC) are directly linked to major soil ecosystem
services such as soil fertility, resistance to erosion, C sequestration and soil $CO_2$ emissions
(Lal, 2004). Understanding such dynamics is therefore of paramount importance for the
challenges of the present century (IPCC, 2014). In particular, the precise quantification of
SOC cycles would allow for a monetization of the respective ecosystem services, and is a
crucial step to overcome the failure of this market (Alexander *et al.*, 2015).
However, the time scale of SOC decomposition, from years to millennia, makes it difficult to
design experiments and requires gathering indirect answers through analysis of monitoring
programs, long-term experiments and SOC turnover models. Most of these models, for





example among the most well-known RothC (Coleman *et al.*, 1997), Century (Parton *et al.*,
1993) and Yasso (Liski *et al.*, 2005), are built around multiple conceptual pools decomposing
with first-order kinetics. This basic structure works well to simulate decadal to centennial
time scales, but shows problems with longer (when considering more protected organic
matter, e.g. Trumbore and Czimczik, 2008) or shorter (when considering microbial dynamics,
e.g. Schimel and Weintraub, 2003) time scales.
Formally, these models could be extended in complexity to represent more accurately all the
processes involved in SOC decomposition that we are aware of. However, a purely
mechanistic modelling approach often fails because the lack of data in respect to the
complexity of the system limits the number of latent variables (all the variables that cannot be
directly measured) that we can infer. A high system complexity, as characterised by multiple
interactions between parameters, causes equifinality problems (Beven, 2006). Representing
such interactions in a way that is both accurate and abstract enough to realistically consider
the availability of data is termed the bias/variance dilemma (Briscoe and Feldman, 2011).
This dilemma represents the most critical point in producing reliable estimates in SOC
modelling.
The struggle of contemporary SOC models becomes more evident when including SO$^{14}$C
data. When time series for both total SOC and SO$^{14}$C are available, they may suggest
contradictory dynamics (Shirato *et al.*, 2013). This confirms the high uncertainty in defining
contemporary SOC model structures and at the same time raises the question of how to use
these two sources of information.
Methods for the inclusion of radiocarbon measurements in SOC models are currently actively
developed. While most SOC models consider $^{14}$C implicitly through the use of mass balance
equations, some attempts have been made to consider $^{14}$C explicitly (Ahrens *et al.*, 2014) as a
separate set of C molecules. A similar approach has been proposed also for $^{13}$C by Ågren *et*
*al.* (1996). The explicit approach offers more flexibility in the representation of processes that
might influence SO$^{14}$C at the price of a minimal increase in model complexity. Nevertheless,
even with explicit consideration of $^{14}$C, modelling results are still not well determined
(Ahrens *et al.*, 2014).
Yet a few studies have considered SO$^{14}$C data within an uncertainty analysis framework.
Braakhekke *et al.* (2014) and Ahrens *et al.* (2014) both considered model uncertainty, but
focused on a single model structure. However, both parameter uncertainty and structural





uncertainty are significant problems endemic to environmental models (Beven, 2002).
Moreover, in both these studies the model sensitivity to radiocarbon was limited to two cases,
either including or excluding $SO^{14}C$ data. The inclusion of $SO^{14}C$ data can modify the model
space substantially (Ahrens *et al.*, 2014) and in a non-linear way. The weight assigned to
$SO^{14}C$ and SOC is a crucial parameter influencing strongly the modelling results, and the
effect of this parameter should, therefore, be studied more in detail.
In order to consider the effect of $^{14}C$ data with respect to structural uncertainty, we calibrated
a set of SOC models over total SOC time series from the ZOFE long-term field experiment
(Oberholzer *et al.*, 2014). In addition, we made use of $SO^{14}C$ measurements in key positions
of the time series. Model structures were built around ICBM, a basic two-pool SOC
decomposition model (Andrén and Kätterer, 1997), and calibrated within a Markov chain
Monte Carlo framework to take care of equifinality and parameter uncertainty. We considered
the possibility of substrate interactions by introducing a control term on decomposition
influenced by the amount of fresh substrate available. To consider the effect of total SOC and
$SO^{14}C$ on the calibration, we assigned a relative weight to the two data streams and calibrated
model structures across a gradient of such weights.
The three research questions driving this work are:
• How will the inclusion of $^{14}C$ data influence the SOC parameters estimated from a multi-
pool model?
• What are the reasons for the observed discrepancy between modelled total SOC and
$SO^{14}C$ dynamics, and which are the most important ones?
• Can we model SOC and $SO^{14}C$ jointly in a way that is minimalistic and flexible and yet
effective?
These research questions generated the following, partially concurrent, hypotheses:
1.    An underestimation of the age of slow C due to the presence of recalcitrant C (e.g.
black C, Leifeld, 2008) or C protected through some other mechanisms is one possible reason
for the observed discrepancy between SOC and $SO^{14}C$ modelled kinetics. Thus, representing
such slow C in the model as inert or particularly slow pool will improve model performances.
2.    An interaction between substrate pools is a process often neglected in C models but
which can contribute the observed discrepancy. Representing this process in the model can
improve model performances.





3. Is it possible to discriminate between the above mentioned processes?
To answer our questions we compared the results from different model structures, each
focusing on slightly different processes. By comparing different model structures we also
aimed at understanding more realistically SOC kinetics in the ZOFE experiments by
acknowledging some model structural uncertainty.
**2    Material and methods**
**2.1    Experimental site**
The data utilized in this study have been collected in the Zürich Organic Fertilization
Experiment (ZOFE, Oberholzer et al. 2014), located in Switzerland at the Agroscope
premises in Reckenholz (Zürich), at 47°25'37" N, 8°31"6' E. The experiment has been
initiated in 1949 and comprises 12 different fertilization treatments, among which we selected
three (Table 1): the control treatment (not receiving any fertilizer input), the mineral
fertilization treatment (receiving yearly 139 N, 28 P, 167 K, 56 kg ha-1 from 1981 and 108 N,
61 P, 318 K, 12 kg ha-1 in the period 1949-1980) and the farmyard manure (FYM) treatment
(receiving yearly 91 N, 24 P, 65 K, 31 kg ha-1 from organic fertilizer and, bi-annually, 1 t
organic carbon from FYM). The site was low-intensity permanent grassland before 1949. Soil
is a Luvisol (WRB, 2007), carbonate-free, with 14% clay, 27% silt and 57% sand. Organic C
content was 1.3% at the beginning of the experiment, and soil pH ($H_2O$) was 6.5. The crop
rotation has a period of 8 years, and includes winter wheat/intercrop-maize-potatoes-winter
wheat/intercrop-maize-summer barley-ley-ley. Main products and by-products of crops are
always removed.
**2.2    Data collection and soil analyses**
The SOC dataset comes from Oberholzer et al. (2014). For modelling, the calibration errors
for both SOC and $SO^{14}C$ has been expressed as coefficient of variation (CV). The CV of the
SOC measurements has been measured independently in 2012 (data not published) and varied
between 0.080 and 0.086 for the different treatments. The $SO^{14}C$ dataset comes from Leifeld
and Mayer (2015). The CV in 2012 varied in this case between 0.017 and 0.029, and has been
extrapolated to the whole $SO^{14}C$ time series. All radiocarbon concentrations utilized here are
expressed in pMC as described in Stuiver and Polach (1977).



In the $SO^{14}C$ time series we assumed that the pre-bomb SOC was at equilibrium with the
atmospheric isotopic value. Although the $SO^{14}C$ might slightly deviate from the $^{14}C$ content
of the atmosphere, the difference between any possible natural discrimination and the effect
of the bomb peak is several orders of magnitude (Goslar *et al.*, 2004) and we regard such a
difference as negligible. In order to improve the calibration of the model in respect to the
$SO^{14}C$ trend, we assumed a fourth $SO^{14}C$ point in year 1955 as corresponding to the
atmospheric signature.
We took the atmospheric $^{14}C$ time series from the Schauinsland station (Levin, Ingeborg and
Kromer 2004; Levin *et al.*, 2013), relatively close to our site (48 km). Radiocarbon values
from May to August are commonly used to represent the vegetation's signature (Levin,
Ingeborg and Kromer 2004), but this implies the assumption of $CO_2$ fixation only in late
spring-summer. We calculated the difference in the time series with and without filtering out
autumn-winter months, after a spline interpolation to regularize the time series, as 3.4 pMC
(root mean squared error), representing a CV between 0.01 and 0.03. This we considered as
negligible and used yearly averages instead.
**2.3   Calculation of C inputs**
The C inputs have been calculated with the C allocation coefficients proposed by Bolinder *et*
*al.* (2007) and in case of potatoes by Walther *et al.* (1994). More details about the input
calculations can be found in Oberholzer *et al.* (2014).
Carbon allocation coefficients may differ between treatments. The potential error introduced
by the nonlinear nature of the root/shoot factor (Bond-Lamberty *et al.*, 2002) was considered
negligible in our case due to conditions being close to optimal for plant growth at our site.
The control treatment still stores as much SOC as treatments with full mineral fertilization
(Oberholzer *et al.*, 2014) and it was still considered to be far from causing extreme deviations
from the selected root/shoot ratio. Another source of error in our estimate is inherent to
extrapolating the original root-shoot relationship (Bolinder *et al*., 2007) to our soil. Such
relationship was built on 168 samples reviewed from the literature of typical agricultural soils,
not different from our alluvial soil, and this error should therefore be small. Another possible
error comes from the lack of estimates for C in form of root exudates.
We considered the above uncertainties for the C allocation by introducing an error factor
calibrated with a uniform prior distribution between 0.8 and 1.2.



## 2.4   Five possible model structures for SOC

The basic model (structure I) is the ICBM model developed by Andrén and Kätterer (1997).
ICBM is a minimalistic model of the general SOC decomposition theory built around two
SOC pools decomposing with first order kinetics. The simplicity of the model allows for a
high degree of flexibility and makes it ideal for model structure explorations, hypotheses
testing and model development.
We used the model stepwise in its recursive form, as derived by Kätterer *et al.* (2004), in
order to follow the highly nonlinear shape of the atmospheric $^{14}$C curve of the last century
(Kurths *et al.*, 1994). The dynamic system representing SOC is described by the following
equations:

$$Y_{(t)} = \left( Y_{(t-1)} + i_{(t-1)} \right) e^{-k_Y r} \qquad (1)$$

$$O_{(t)} = \left( O_{(t-1)} + \varphi_{Y(t-1)} \right) e^{-k_O r} + \varphi_{Y(t-1)} e^{-k_Y r} \qquad (2)$$

$$\varphi_{Y(t-1)} = h_1 \frac{k_Y \left( Y_{(t-1)} + i_{(t-1)} \right)}{k_O + k_Y} \qquad (3)$$

The SOC at time *t* is therefore calculated as:

$$Tot_{(t)} = Y_{(t)} + O_{(t)} \qquad (4)$$

This system describes the evolution of two C pools, young (*Y)* and old (*O)* SOC,
decomposing with rate $k_Y$ and $k_O$. Their mean residence time (MRT) is defined by the
reciprocal of the decomposition constants, or $\dfrac{1}{k_Y}$ and $\dfrac{1}{k_O}$. The term φ describes the flux
between the two pools, which is controlled by the humification coefficient $h_1$ that defines the
amount of carbon that goes from *Y* to *O*. The term *r* aggregates climatic and edaphic
influence, and is calculated according to equations that follow in the text. The system of Eq.
(1), (2), (3) and (4) can then be modified in order to represent different hypotheses. The
model defined by the system of Eq. (1), (2), (3) and (4) is therefore calibrated for 4 unknown
parameters, namely $k_Y$, $k_O$, $h_1$ and the initial distribution of C between pools *Y* and *O*.



A first modification (i.e. model structure II), already suggested by Juston (2012), adds a static
pool representing SOC cycling at extremely slow decomposition rates. This pool is virtually
inert and does not interact with the other pools or decomposes. Since the SOC age spectrum is
likely distributed according to a logarithmic function of age (Bosatta and Ågren, 1999), this
approximation may be reasonable for very slow SOC atoms. Eq. (4) can therefore be modified
by adding an "inert" pool $R$ as:

$$Tot_{(t)} = Y_{(t)} + O_{(t)} + R \qquad (5)$$

This modification adds one parameter to the initial calibration to represent the initial value of
$R$.
A second modification, i.e. model structure III, introduces instead of a static third pool a
decomposing third pool. The dynamics of the $R$ pool in Eq. (5) now are similar to $O$ in Eq.

12      (2):

$$R_{(t)} = \left( R_{(t-1)} + \varphi_{O(t-1)} \right) e^{-k_R r} + \varphi_{O(t-1)} e^{-k_O r} \qquad (6)$$

$$\varphi_{O(t-1)} = h_2 \frac{k_O \left( O_{(t-1)} + \varphi_{Y(t-1)} \right)}{k_R + k_O} \qquad (7)$$

This modification adds two more unknown parameters to the initial model, namely $k_R$ and $h_2$
(table 2).
A third modification of structure I, i.e. model structure IV, modifies the basic set of equations
with a single, aggregated term to account for the effect of "young" substrates on microbial
dynamics and therefore on decomposition rates. We modified Eq. (1) and (2) by adding a
term α in the exponent of the decomposition function according to Wutzler and Reichstein
(2013). Since the fluxes from the slower and older pool are small compared to the flux from
the younger pool we approximated the system by neglecting the former in calculating α as
already suggested by Wutzler and Reichstein (2013). The resulting equation defining α is:

$$\alpha_{(t)} = \max\left( 0,1 - \frac{\beta}{k_Y \left( Y_{(t)} + i_{(t)} \right)} \right) \qquad (8)$$

where $\beta$ represents a lumped term aggregating microbial limitations on decomposition
(Wutzler and Reichstein 2013). The term α is introduced as a modifier for both $k_Y$ and $k_O$.
The denominator represents the maximum possible microbial uptake, which is the total flux





from $Y$ to $O$. When the flux from the young pool is below the value of $\beta$ decomposition goes
to zero, but when this flux increases above this value decomposition approaches $k_Y$ and $k_O$.
This model structure adds one more unknown parameter (Table 2). Finally, model structure II
was extended by a substrate control as in structure IV to give structure V. All model
structures were run in annual time steps.
For model structures III and IV, with a substrate interaction term, an alternative MRT could
be defined as $\dfrac{1}{k \cdot \alpha}$. Although, since its discussion goes beyond the scope of this manuscript,
we did not consider such definition for our results, we reported it in order to better explain the
numerical effect of Eq. (8) on MRT.

## 2.5 Model structure for SO$^{14}$C

Each model structure was extended by running a separate system of equations for SO$^{14}$C.
With the introduction of SO$^{14}$C, the number of parameters increases (Table 2). We calculated
the ratio of $^{12}$C$^{/14}$C from the pMC value according to the definitions given in Stuiver and
Polach (1977), and calculated from this ratio the mass of $^{14}$C. We set the $\delta^{13}$C normalization
factor at -26‰, close to that of a typical C3 soil. Most parameters were assumed to be the
same as for SOC except for the initial distribution of the SO$^{14}$C pools which was allowed to
vary by using a normal prior distribution centered on the mean of SOC pools distribution and
with a coefficient of variation of 0.1.
The radiocarbon decay is considered by adding the term λ, corresponding to $\dfrac{1}{8265}$ yr$^{-1}$
(Stuiver and Polach 1977), to all decomposition constants which then become $k_{pool} + \lambda$.
We did not consider a time lag between C assimilation and release into the SOC cycle
because we are considering an agricultural system with annual plants. These plants have a
physiological time lag of few hours (Kuzyakov and Gavrichkova, 2010) and eventual storage
compounds are released at the end of the cultural cycle, which is in most cases less than one
year. The years during rotation where leys are present are few (Oberholzer *et al.* 2014). With
the annual resolution utilized in this study the time lag could therefore considered being
negligible.





The effect of the two data streams (SOC and SO$^{14}$C) on the calibration of the model structures
has been tested by introducing an arbitrary weighting term. This value, between 0 and 1, acts
in the Bayesian calibration to modify the variance of the probability distributions representing
the two time series. When the weighting term tends to one, the variance defining the SOC
probability distribution tends to zero while for the SO$^{14}$C probability distribution it tends to
infinite (S1). This alters the weight of that particular time series on the joint posterior
distribution of the calibrated values. The precision of the SO$^{14}$C data stream tends to zero and
so it does not influence the calibration. When the weighting tends to zero, the opposite
applies.
In order to better capture the effect of adding the information contained in the SO$^{14}$C data
stream in the calibration, we run all the calibrations over a gradient of such weights (with
discrete values 0.05, 0.175, 0.350, 0.500, 0.650, 0.825, 0.950).
Since the two data streams are not homogeneous, this weighting term is considered as an
empirical evaluation of the sensitivity of the model. It is an effective method for assessing the
relative effect of the information from either isotope and offers more detail compared to
testing only for the two options (SOC only and SOC + SO$^{14}$C) separately.

## 2.6    Considering kinetic isotope effects in soil

A possible differential loss of SO$^{14}$C compared to SOC, caused by kinetic isotope effects
(Tsai and Hu, 2013), is accounted for by the standard normalization of $^{14}$C values for $\delta^{13}$C.
Since every process that possibly causes a variation of the $^{13}$C content from the moment that
the $CO_2$ was fixed might be assumed squared on $^{14}$C (Stuiver and Polach 1977), the
normalization considers any process that can influence the C signature. This normalization
relies on the assumption that the $^{13}$C/$^{14}$C ratio in nature is stable, since every molecule
originates from atmospheric $CO_2$ which is supposedly homogeneous in open air. The Suess
effect, a change in the atmospheric isotopic composition triggered by the burning of fossil
fuels (e.g. Francey *et al*., 1999), does not represent in this sense a problem since the $^{14}$C
values are calibrated over atmospheric time series. Errors in the correction might be
introduced by eventual local hot spots (e.g. industrial contaminations) for the atmospheric
$^{13}$C/$^{14}$C ratio. Our site, located at few kilometers from any major industry and hundreds of
meters from any building, should be relatively free from local contamination sources and the
closeness of the site to the measurement of atmospheric $^{14}$C time series should account for



regional variations. Nevertheless, we considered the possible error associated with these
assumptions by allowing the initial ratios of the [14]C pools to vary slightly for [14]C by assigning
a normal prior distribution to them, centered on the SOC ratios with deviation corresponding
to 1% of these values.
**2.7   Climatic and edaphic variables**
The parameter $r$ in Eq. (1) and (2) in the original ICBM calibration (Andrén and Kätterer,
1997) aggregates all the influences on SOC from soil type and climate. It was originally
conceived as a constant, but it has been used also as a response variable connected with
climatic and edaphic factors (Andrén *et al.*, 2012). We decided to consider $r$ according to the
following equation:
$$r_{(t)} = r_{Temp(t)} \cdot r_{Moist(t)} \cdot \varepsilon \qquad (9)$$
where $r_{Temp}$ and $r_{Moist}$ are the decomposition rate modifiers due to temperature and soil
moisture, respectively and $\varepsilon$ is an error term.
In this particular case we included proxies for soil temperature and soil moisture and we
selected the two climatic functions from the CENTURY model (Parton *et al.*, 2001; Bauer *et*
*al.*, 2008), since they adapted well to the data available for this experiment. The temperature
function was adopted as following:
$$r_{Temp(T)} = 0.560 + 0.465 \cdot \arctan\left(0.097 \cdot (T - 15 - 7)\right) \qquad (10)$$
while the moisture function was adopted as following:
$$r_{Moist(\theta)} = \left(1 + 30 e^{\left(\frac{-8.5 \cdot PPT}{PET}\right)}\right)^{-1} \qquad (11)$$
where $T$ is soil temperature (° C), $PPT$ is the sum of stored water and precipitation, in our case
approximated to total accumulated precipitation for the reference period due to the nature of
our dataset and $PET$ is the potential evapotranspiration (Primault, 1962). The term $\varepsilon$ has been
described with a uniform distribution between -0.5 and +0.5.





Meteorological data were obtained from the Swiss Federal Research Station for Agroecology
and Agriculture Zürich-Reckenholz (FAL), located at less than 100 m from the ZOFE
experiment.
In order to maintain comparability of results with the original ICBM model, $r$ has been
normalized with its mean value as $r_{norm(t)} = \dfrac{r_{(t)}}{\overline{r}}$, therefore making it vary around 1. The
normalization, together with the introduction of the $\varepsilon$ term in Eq. (9), reconciles the climatic
functions with ICBM. The resulting variation of the $r_{norm}$ term is pictured in S3. Since we are
comparing three treatments in the same field we do not need to take into account any
difference in climate between the plots, and we can use the climatic parameter only to account
for variability in the data that might be due to inter-annual climatic variation.
**2.8   Model calibration, initialization and prior assumptions**
Given the close interactions between the kinetic parameters a deterministic optimization
algorithm might miss possible equifinality (Beven, 2008). We therefore relied on a
Metropolis-Hastings algorithm (in the implementation of JAGS, Plummer 2003), with
likelihood function according to a formal Bayesian statistical framework.
We assumed that the parameters defining the SOC pools (namely $k_{pool}$, $h_{pool}$ and the initial
pool distribution) were the same for all treatments. Every calibration has been run in 4
separated Markov chains, and the convergence of the chains has been assessed visually
through the use of Gelman's plots (Brooks and Gelman, 1998). Each chain was calibrated
with a first adaptation period of 10.000 runs of which 5000 have been discarded as burn-in
period, and then 100.000 search runs. The chains always showed reasonable convergence.
Priors for the rates ($k_{pool}$) have been considered as normally distributed, with mean value
coming from Andrén and Kätterer (1997) and deviation set to half of the mean value. The
mean of the prior for $k_O$ has been set considering it as a fixed ratio of the value of $k_Y$. Also
this ratio (0.075) has been calculated from Andrén and Kätterer (1997). The priors for $h_Y$
have been considered normally distributed. Mean values to represent the different input
qualities were calculated as averages of all the scenarios reported in Kätterer *et al.* (2011) as
following. By assuming the composition of the young pool being similar to the inputs, we





chose the prior value for $h_Y$ for the control and the mineral fertilizer treatments as 0.185
(which is the average for roots and shoots) while for the farmyard manure the chosen value
was 0.265. We have chosen for this parameter stronger prior distributions by setting its
deviation to 10% of the mean value. In the third model structure the $h_O$ prior has been set as
an uniform distribution between 0 and $h_Y$.
Priors for the initial distribution of the SOC pools were considered uniformly distributed
between 0 and 100% of initial SOC but constrained by the mass balance, i.e., the sum of SOC
mass in all pools should add up to 100% of initial SOC. Priors for the initial distribution of
the pools for $SO^{14}C$ were generated with a uniform distribution using the portion of total SOC
pools as mean and variance set to 1% of this value.
**2.9   Model comparison and selection**
Following the same principle of simplicity maximization on which we built the whole study,
we selected the Akaike information criterion (AIC) to estimate the information content of the
model structures. The AIC has been calculated as:

$$AIC = 2p + n \cdot \log\left(\frac{RSS}{n}\right) \qquad (12)$$

where $p$ is the number of parameters, $n$ is the number of samples and $RSS$ is the residual sum
of square of the model.
The use of the $RSS$ in Eq. (12) is a simplification, since it is a metric only proportional to the
likelihood. The difference lies in the lack of one integration constant. Since the AIC is used in
this study only for a relative comparison between model structures, we considered this
approximation justifiable. The use of the AIC rather than RMSE for measuring model
performances can capture how the different model structures react to the introduction of the
additional stream of information, i.e. $SO^{14}C$, by acting as a structure-dependent normalization,
allowing for a performance comparison between different structures. Also the best weighting
parameter representing the partial weight of SOC and $SO^{14}C$ data has been selected according
to the smallest AIC.
The choice of the AIC is motivated by its simplicity (explicit also in the intention of his
author, Akaike, 1974), and by the consideration that we are comparing models over exactly




the same number of samples (Burnham and Anderson, 2004). But since the choice of any
model performance indicator is highly subjective, we also calculated for all the models the
deviance information criterion (DIC, Plummer, 2008) for comparison with the AIC.
**3   Results**
**3.1   Effect of the SOC data stream on model performances**
In general the addition of the $SO^{14}C$ data always improved the performance of the calibrations
until a certain optimal point. This effect was similar for any of the different model structures,
and an eventual relative advantage of one structure above another in considering information
from $SO^{14}C$ data was not evident. The improvement increased for every structure up to a
partial weight of 0.35, and then worsened marginally when moving forward toward a higher
weight of $SO^{14}C$ data (Fig. 1). However, the decrease in performances was dramatic when
moving towards a bigger relative weight of $SO^{14}C$ data.
The introduction of the $SO^{14}C$ data stream in general decreased the uncertainty of the
parameters until an optimal weight for all the models without a substrate interaction
(structures I, II and III), and the average coefficient of variation of the parameters followed a
general pattern similar to the average AIC (S2). For the structures including substrate
interaction (VI and V) the pattern was oscillating in a more complicated way, making it
impossible to identify any consistent trend. The RMSE (Fig. 2) of the model structures was
closely related to the AIC but with different relative values for the different structures.
**3.2   Optimal model choice**
Overall, the "best" model structure indicated by the AIC to best describe our data was the
basic ICBM, structure I (Fig. 1). This is particularly true for the FYM treatment (with highest
SOC), which was the treatment best described by all our model structures.
The average RMSE was similar for all model structures, but there were small differences.
Unexpectedly, structure III did not present the lowest average RMSE among all structures
(Fig. 2), although it has the highest number of parameters. Structure II was the one which
performed the best in terms of RMSE.




We compared these five structures also through DIC, which was 591.9 for structure I, 579.9
for structure II, 593.8 for structure III, 603.1 for structure IV and 591.9 for structure V. Also
the DIC indicated better performances of simpler structures and it indicated structure II as the
best model. However, it did not indicate any difference between the second and third best
choice (structure I and V) and differences were not as evident as when using AIC.

### 3.3   SOC distribution and kinetics in the ZOFE experiment as estimated by different model structures

The MRT (Fig. 8) of the old pool, according to structures I and II, were 94.99±0.10 and
78.93±0.11 years, respectively, while the ones for the young pool were 5.91±0.09 and
5.33±0.08years, respectively. Owing to the introduction of an additional term, modifying the
kinetic in relation to the amount of young substrate, the results differ for structures IV and V.
Here, MRT results were 14.87±0.85 and 16.76±0.45 years for the old pool and 0.85±0.34 and
1.01±0.30 years for the young pool, respectively. Structure III determined pool definitions
similar to structure I and II; and in this case the MRT was 98.85±0.10 years for the old and
4.22±0.10 years for the young pool. The third, "recalcitrant" pool in structure III revealed a
MRT of 477.78±0.66 years. Simulation results are shown only for structure I (Fig. 6) and II
(Fig. 7), and for structure II, III and V in S5, S6 and S7.
The estimated size of the initial pools did not vary much among the selected model structures
(Fig. 9). The amount of carbon in the young pool ranged from 15.37±1.64 Mg ha$^{-1}$ (structure
I) to 11.37±1.50 Mg ha$^{-1}$ (structure III). The amount of carbon in the old pool ranged from
22.70±1.59 Mg ha$^{-1}$ (structure I) to 20.28±1.74 Mg ha$^{-1}$ (structure IV) for structures
considering only two pools, while it ranged from 25.25±1.39 Mg ha$^{-1}$ (structure II) to
23.00±1.70 Mg ha$^{-1}$ (structure III) for structures considering three pools. As evident from
Figs. 3, 4 and 5, these results are also strongly dependent on the choice of the weighting
parameter between the SOC and the SO$^{14}$C data streams.
All the tested model structures, and within all the tested values of the weighting parameter,
inferred a change right after the land use change in the ZOFE trial. In all treatments without
amendments, the young pool decreased rapidly within a few years after conversion from
grassland to FYM and mineral fertilization. In structures I this decrease was more dramatic,
while more complex models (II, III, IV and V) could describe the observed trends as more
gradual thanks to the additional number of parameters.





**4  Discussion**
**4.1  Effect of the C data stream on the kinetics of SOC pools**
During calibration all model structures seemed to react to the SO$^{14}$C data by reducing
decomposition rates and humification coefficients, i.e., the introduction of SO$^{14}$C decelerated
the simulated C dynamics. For structure I the effect of adding the SO$^{14}$C data seemed to slow
down the decomposition of both pools (Fig. 3). This decrease was associated with a decrease
of the humification coefficient, hence reducing also the flux of material that goes from a faster
to a slower pool. In the same time the relative size of the slower pool decreased. For structure
IV (Fig. 3) the addition of a substrate interaction term made the decrease in speed associated
with the introduction of SO$^{14}$C data more dramatic and in some specific cases more difficult
to interpret, but in general following a similar trend. In structures with a third inert pool, II
and V (Fig. 4), trends were replicating those with only two pools. Structure V presented a
pattern very similar to structure IV. The inert pool proportion increased with the increase of
the weight of SO$^{14}$C data. Also results from structure III (S5) indicate a consistent reduction
in the speed of C cycling with the introduction of the SO$^{14}$C data in every parameter. In
general we can affirm that the inclusion of the SO$^{14}$C data decreased the size of the slower O
pool while it increased the residence time of both Y and O pools.
None of our tested model structures could represent consistently both data streams at the same
time. For the SO$^{14}$C value measured in 1973, every model structure under-predicted the
isotopic value of SOC particularly for the low input treatment. Conversely, the last SO$^{14}$C
point, measured in 2012, was consistently over-predicted by every model structure. This
suggests that all our model structures are still failing to represent some key process related to
SOC decomposition.
The use of the radiocarbon bomb peak to constrain SOC turnover models, although in use
since decades (Trumbore, 1989), has often raised similar controversies. The implicit inclusion
of $^{14}$C data in C models through mass balance functions produced discrepancies between
modelled and measured values in a recent study by Shirato *et al.* (2013). In another study
(Rethemeyer *et al.*, 2007) this approach was judged as a viable option. The explicit
consideration of $^{14}$C pools did not offer in this sense any advantage over implicit models.
Braakhekke *et al.* (2014), using a soil profile model, found that the addition of SO$^{14}$C data as
new constrain produced an increase in the uncertainty of the SOC stocks in the individual



layers, while improved just marginally the total SOC stock estimate. Ahrens *et al*. (2014)
utilized SO$^{14}$C data to constrain an isotopically explicit single layer model in a situation
where data about SOC kinetics were scarce. In that case the problem of model initialization
was partially solved with additional information coming from $^{14}$C, but the high uncertainty of
the considered system did not make it possible to determine if one site was losing or gaining
carbon, and the strong interaction between MRT and deviation from the steady state made
evident a trade-off between estimates with and without using SO$^{14}$C data.
One of the possible reasons for the recorded discrepancies in the estimates from models
conditioned with and without SO$^{14}$C data might be the absence of microbial dynamics in SOC
stabilization (Riley *et al.*, 2014). Ahrens *et al*. (2015), with a rather mechanistic model,
recently suggested that a control on biologically mediated depolymerization can explain alone
some of the observed discrepancies. But the performances of structure IV and V on our
dataset, lower in terms of AIC compared to the simpler structures I and II, did not allow us to
confirm such a hypothesis. Another possible explanation for the discrepancy between models
and measurements is the presence of recalcitrant and old organic carbon not well captured by
our model structures. Structure II was selected by the AIC, while structure III, although not
performing best with AIC due to the high number of parameters, presented a good RMSE.
Compared to the basic structure I both these structures introduced an additional slow SOC
pool. Some form of chemical recalcitrance cannot therefore yet be ruled out.
In our study we focused on the optimal utilization of the information contained in SO$^{14}$C data
together with the minimization of model complexity. We found a relevant improvement of the
overall model performances when also SO$^{14}$C data were introduced but only until an optimal
weight, while beyond that weight model performances decreased substantially. It is difficult
to generalize our optimum as a general recommendation since it also depends on the density
of the two data streams, but our results suggest that the relative weight of the two
measurements is an additional parameter that must be considered and optimized whenever the
SO$^{14}$C data are used for model constraining.
A generalizable and detailed mechanistic understanding of SOC stabilization is not yet
available, and SOC models are still facing a deep parametrical and structural uncertainty.
According to some authors (e.g. Beven, 2002) such uncertainty is inherent to the nature of
ecosystem modelling, and needs to be accepted and considered in developing new





methodologies. In this perspective we adopted a pragmatic approach to determine the optimal
weighting factor, which turned out to be a crucial step with large impact on modelling results.

## 4.2  SOC dynamics in the ZOFE experiment as estimated by different model structures

All the model structures indicated a rapid decrease in the young pool following the conversion
from grassland to cropland. This means that the annual inputs under the new management
were too small to replenish the C in the former young pool while most of the material is either
decomposed or humified in the old pool. This is not unlikely since also by-products, like
straw, are removed, and the inputs from the cropland management are greatly reduced
compared to a low-intensity grassland (Rumpel *et al*., 2015), where a lot of the net primary
productivity is either retained or returned in form of excrements. Furthermore, the disruption
of the soil structure that formed under permanent grassland caused by the conversion may
have released and subsequently mineralized largely undecomposed organic matter, such as
particle or light fractions previously protected inside aggregates (Six and Paustian, 2014).
After this re-equilibration of the young pool, the slower but constant decrease in the total SOC
was explained by all the models with a slow but constant decrease in the old pool, missing the
inputs previously received from a bigger young pool. All our model structures indicated that
the considered treatments in the ZOFE experiment are all still far from a new SOC
equilibrium.
The error in the simulated $SO^{14}C$ might be due to an overestimation of the speed of the C
cycle. Nevertheless the fact that more complex model structures (IV, V and III) did not
present any advantage over simpler (I and II) structures makes it difficult to judge the weight
of the two represented processes (stabilization of SOC, represented by an additional "inert"
pool, or substrate feedbacks. The same discrepancy in predictions might also be caused by a
systematic underestimation of the inputs. Except for the highest input treatment (FYM), the
posterior probability distribution for the assumed input error term (S4) was always skewed
toward the upper limit. This suggests some kind of systematic error concentrated in the lower
end of the input range. Hence, the application of linear allometric functions to estimate carbon
inputs from yields, as adopted here, must be treated with caution. The relatively symmetric
distribution (and in general lower value) of the input error term for the FYM treatment in



structures I, II and III points out that model structures not considering substrate interactions
might be more robust in cases of input uncertainty.
Another possible reason for the error in model predictions might be the nature of the error in
the SO$^{14}$C series. This has been estimated by Leifeld and Mayer (2015) from the last time
point and subsequently extrapolated to the whole time series, assuming therefore normality
and homoscedasticity over time. These assumptions might not always hold in soil systems,
and this would be particularly crucial in the case of the 1973 point in the control treatment.
Further investigation, focused in particular to the belowground production in the ZOFE
experiment, is needed for determining the reasons for such error.

### 11    4.3    Initial SOC distribution and MRT of SOC pools in the ZOFE experiment as
### 12             estimated by different model structures

Our results for the kinetic parameters are in general in the same order of magnitude than what
was reported in the literature (Andrén and Kätterer, 1997), although the introduction of the
SO$^{14}$C forced a deceleration of the C cycle.
The estimation of MRT strongly depends on all the assumptions in the model structure, and
the high uncertainty around what might be the "best" structure is pointed out by the
disagreement of the different criteria used for selection, which highlights the fact that there is
no true model (or that "all models are wrong", Box, 1976). The combination of several
structures, although difficult to perform in practice (Refsgaard *et al.*, 2006), might therefore
represent a reasonable option and deserves further attention.
The MRT estimates (Fig. 8) depend on the introduction of a substrate control term in the
model structure, but once this was accounted for it seemed quite robust. We must consider
here that the introduction of a substrate control term as described by Eq. (8) modifies the
definition of the decomposition constants, and therefore the MRT calculated accordingly.
When introducing also the term $\alpha$ in the calculation of MRT this ranged between 2.78 and
3.13 and 46.00 and 54.47 years for young and old pool respectively, so not far from what
indicated by the other structures. A detailed discussion about the MRT definition is outside
the scope of this study, but here we want to make clear that a direct comparison of the MRT
between these two groups of structures according to a common definition would not be
meaningful and the differences in the model structure must be accounted for.





Model initialization seemed quite robust, with values substantially not differing between
models with the same number of pools.

## 4.4  Balancing the bias/variance dilemma in SOC modelling

As suggested by the multiple structures evaluated in this study, the conceptual nature of SOC
pools makes their definition volatile. Each pool is a theoretical construction defined
specifically by assumptions at the level of model structure as well as model calibration.
Some attempts have been made to reconcile a definition of C pools with real measurements.
For example the well-established forest model Yasso (Liski *et al*., 2005) bases its calibration
on data from chemical litter fractionation, which gives the initialization values for the
different C pools. But the fractionation behind Yasso might seem questionable in agricultural
soils where inputs are often homogenized with the mineral fraction and less, if at all,
identifiable. In more homogenized mineral topsoils the main obstacle to this approach is that
available fractionation methods do not reflect precise stabilization processes (von Lützow *et*
*al.*, 2007). One of the most promising recent attempts to develop a non-theoretical
quantification of SOC pools in agricultural/mineral soils is the one by Zimmermann *et al*.
(2007), which tried to develop a measurement standard for RothC (Coleman *et al.*, 1997)
pools. All these methods share in common the risk that correlations between the
measurements and the theoretical pools might be strongly localized (or difficult to reproduce,
Poeplau *et al.*, 2013). This is not surprising given the complexity of SOC stabilization
mechanisms (Kleber *et al.*, 2011). Indications are that stability should be considered as an
intrinsic property of the soil ecosystem (Schmidt *et al.*, 2011) and thus local. It is therefore
problematic to generalize a fractionation methodology that reflects in detail SOC stabilization
processes, which would in turn define SOC pools.
Hence, we still need to aggregate the available information in a theory of SOC decomposition
that is simple enough to be generalizable. This way the model structure represents the SOC
decomposition processes in an aggregated (and simplified) way that is compatible with the
amount of knowledge at disposal. The challenge of conciliating predictive power, and
therefore practical value of our models, with accuracy is the formulation of the bias/variance
trade-off as found in modern soil science.





As suggested from our dataset, which although not perfect is already relatively rich in
information and not far from the best possible conditions available for soil carbon modelling,
the information available for inverse modelling discrimination still seems insufficient to
validate models that are too mechanistic.
## 5 Conclusions
The SOC in the ZOFE experiment underwent a profound decrease after the initial land use
change from grass- to cropland. This decrease was described in the first years by all our
model structures as a fast re-equilibration of the young pool, which decreased rapidly after a
reduction of the inputs and/or an increased mineralization and caused in consequence a slower
but constant decrease in the older pools. In the long term, treatments not receiving organic
fertilization were still losing C even more than 60 years after land use change. The estimates
of the MRT in the ZOFE experiment were robust once accounted for differences inherent to
the model structures. Comparable model structures (in particular I, II and II) were relatively in
agreement, and the influence of the number of pools on MRT was instead quite limited.
The introduction of $SO^{14}C$ data during calibration improved performances of all model
structures and reduced the uncertainty of the parametrization. It also made clear the existence
of a trade-off between representing the information from $SO^{14}C$ and SOC when utilizing a
multi-pool SOC model structure. None of our five structures seemed able to reconcile
consistently the two data streams. This suggests the presence of processes that were implicit
in the $SO^{14}C$ data stream but not well described in our model structures, which caused the
information from the $SO^{14}C$ to have a strong impact on the results. We therefore suggest the
explicit consideration of a weight associated with each data stream as a routine procedure
whenever $SO^{14}C$ data are considered as an additional model constrain.
In our data set, the best model performances were achieved by the two simpler models,
pointing out that the data available do not allow for a more detailed mechanistic SOC
modelling. Although processes based on interactions of part of the substrate with the
decomposition kinetics might explain the observations, recalcitrance inherent to the substrate
(corresponding to the adoption of a slower additional decomposing C pool) remains a valid
alternative in explaining the data.



1  **6   Data availability**

2   All the data on which this study is based are published in previous studies and the sources are

3   cited in the text.



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



Table 1: The treatments considered in this study. $^\dagger$= kg ha$^{-1}$ y$^{-1}$, $^{\dagger\dagger}$= Mg ha$^{-1}$ y$^{-1}$, ♣=from
organic amendment. [a]=1949-1980, [b]=since 1981, [c]=1949-1990, [d]=since 1991 *=average.

| Treatment | Annual input | | | | | | Initial SOC$^{\dagger\dagger}$ | Final SOC$^{\dagger\dagger}$ |
|---|---|---|---|---|---|---|---|---|
| | N$^\dagger$ | P$^\dagger$ | K$^\dagger$ | Mg$^\dagger$ | Fertilizer C$^\dagger$ | Estimated total C$^\dagger$ | | |
| Control | 0 | 0 | 0 | 0 | 0 | 580 | 38.75 | 24.28 |
| N2P2K2Mg | 108[a]/139[b] | 61[c]/38[d] | 318[c]/167[d] | 12[a]/56[b] | 0 | 1350 | 38.75 | 27.05 |
| Farmyard Manure | 91♣ | 24♣ | 65♣ | 31 | 2500 | 3621 | 38.75 | 31.70 |

Table 2: Summary of the model structures tested in this study (considered here in their basic
forms for total C only and for the two isotopes together.

| | Struct. I | Struct. II | Struct. III | Struct. IV | Struct. V |
|---|---|---|---|---|---|
| Description | Two pools | Two pools + Inert | Three pools | Two pools + substrate control | Two pools + substrate control + Inert |
| Parameters (SOC) | 4 | 5 | 7 | 5 | 6 |
| Parameters (SOC+SO$^{14}$C) | 4+1 | 5+2 | 7+3 | 5+1 | 6+2 |





4    Figure 1: Average of the AIC among all the three treatments for the five model structures with the variation of

5    the relative weight of SO$^{14}$C over total C. In this scale 1 means only total C, 0 means only SO$^{14}$C.



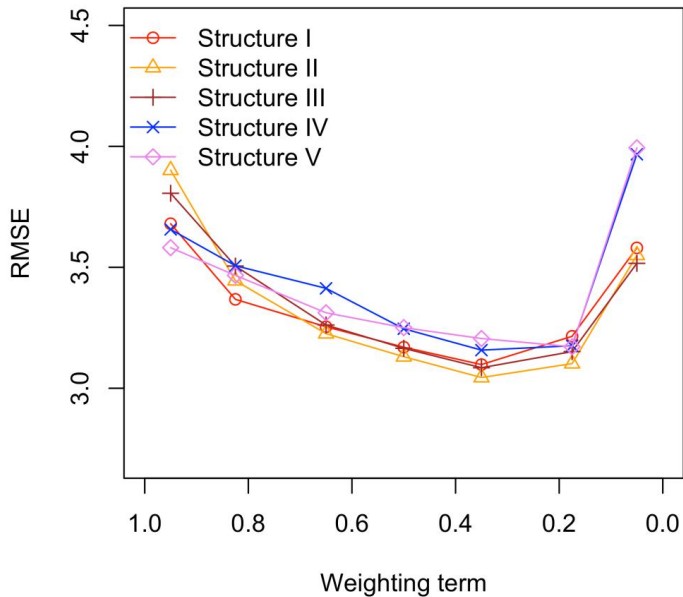

4   Figure 2: Average of the RMSE among all the three treatments for the five model structures with the variation of

5   the relative weight of $SO^{14}C$ over total C. In this scale 1 means only total C, 0 means only $SO^{14}C$ .



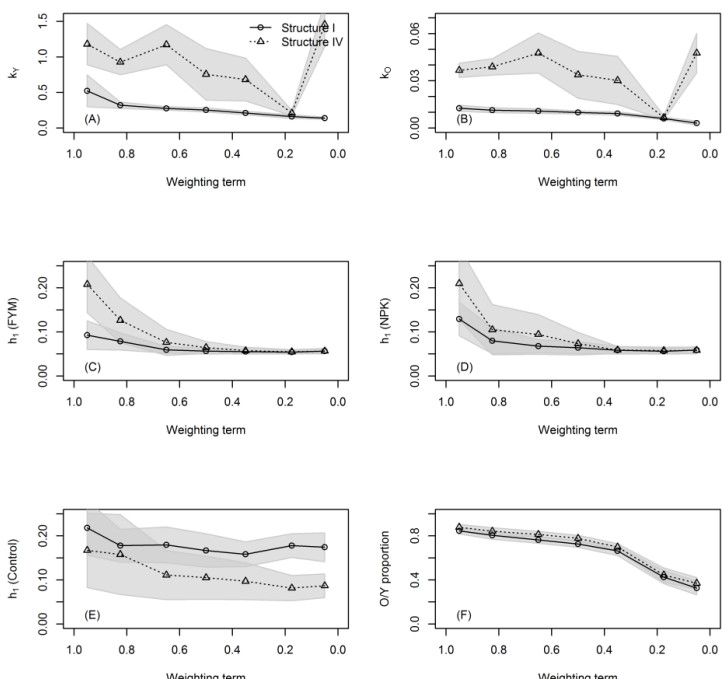

Figure 3: Effect of the SO$^{14}$C stream over the main SOC parameters in structures I and IV. In this scale 1 means only total C, 0 means only SO$^{14}$C. The shaded areas represent the error of the calibrated parameter (calculated as standard deviation of the whole Markov chain).



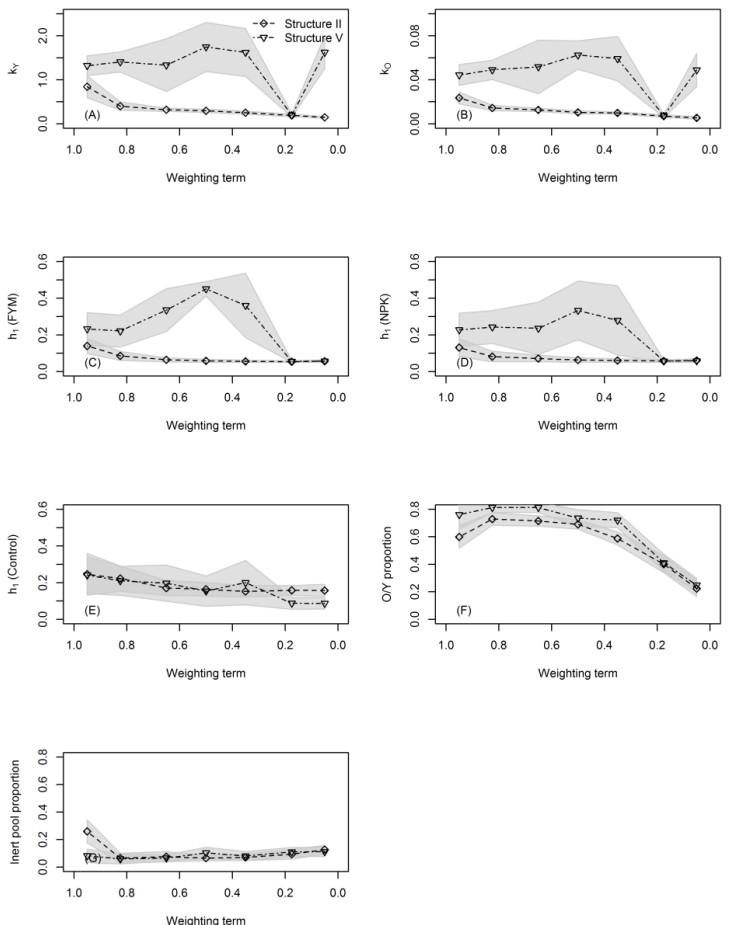

2                                                                                    Figure 4: Effect of the
3   SO$^{14}$C data over the main SOC parameters in structure II and V. In this scale 1 means only total C, 0 means only
4   SO$^{14}$C. The shaded areas represent the error of the calibrated parameter (calculated as standard deviation of the
5   whole Markov chain).





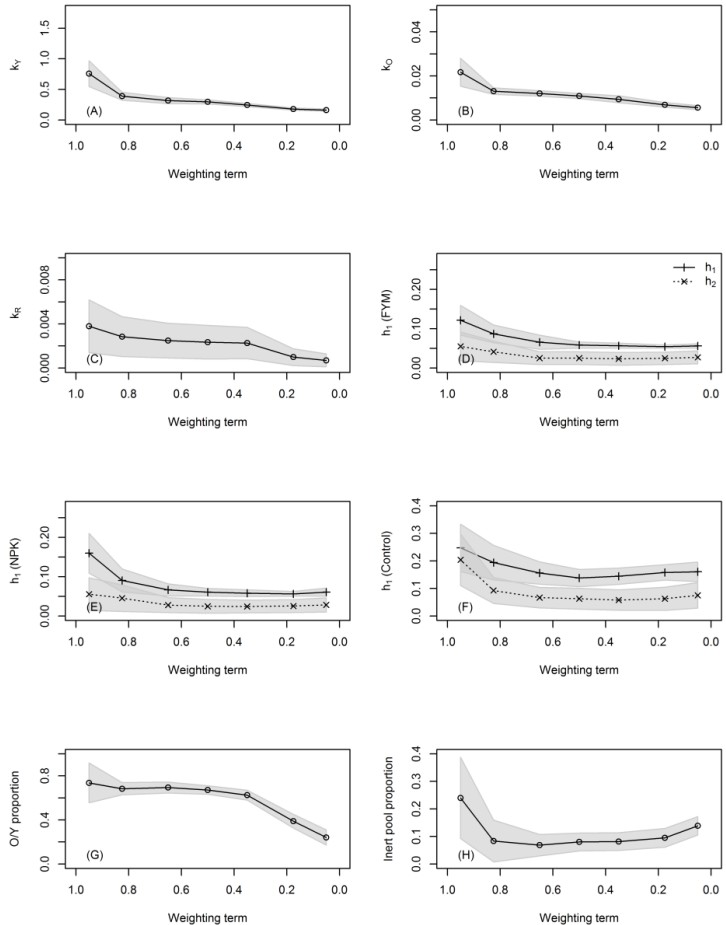

Figure 5: Effect of the SO$^{14}$C data over the main SOC parameters in structure III. In this scale 1 means only total C, 0 means only SO$^{14}$C. The shaded areas represent the error of the calibrated parameter (calculated as standard deviation of the whole Markov chain).





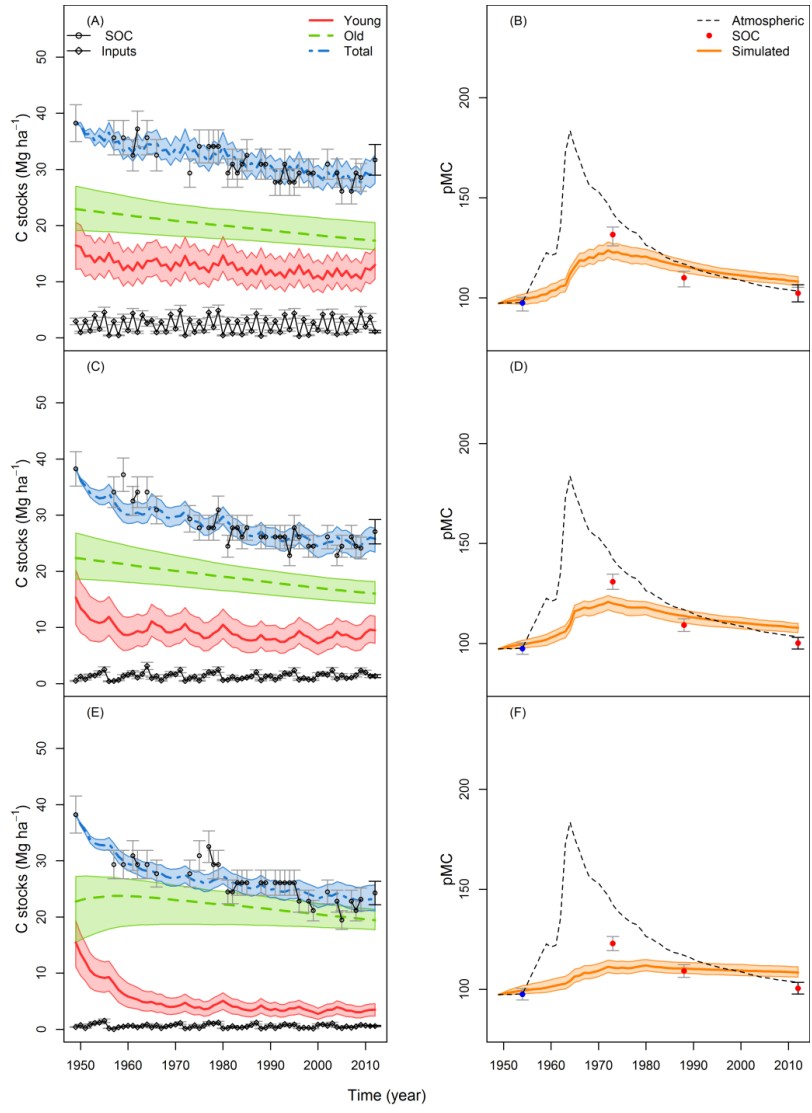

2 Figure 6: Simulation of SOC pools in the ZOFE trial as described by model structure I, with weighting factor =

3 0.35. Error bars represent the measured (black) and estimated (dark grey) standard error of the measurements.

4 SOC (A,C,E) is in Mg ha$^{-1}$, while SO$^{14}$C (B, D, F) is in pMC.





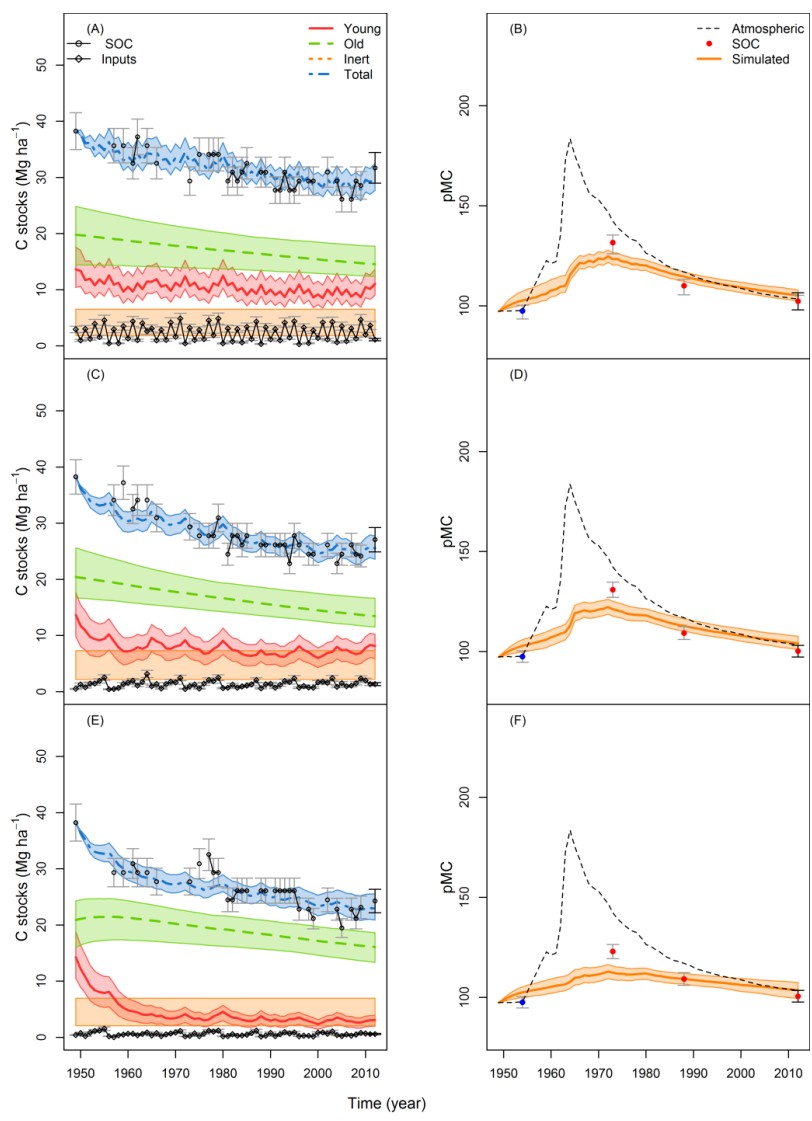

Figure 7: Simulation of SOC pools in the ZOFE trial as described by model structure II, with weighting factor = 0.35. Error bars represent the measured (black) and estimated (dark grey) standard error of the measurements. SOC (A,C,E) is in Mg ha$^{-1}$, while SO$^{14}$C (B, D, F) is in pMC.





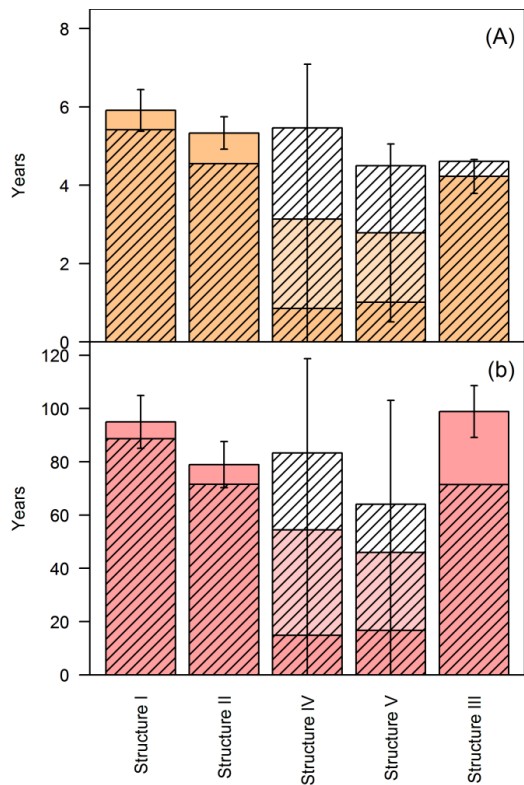

Figure 8: MRT of the young pool (A) and old pool (B) of SOC in the ZOFE trial as indicated by the model
structures examined, with weighting factor = 0.35 (solid colored area) and weighting factor = 0.65 (shaded area).
The solid lighter colored area denotes the MRT calculated (for structures IV and V) according to $\frac{1}{k \cdot \alpha}$, while
the darker colored area according to $\frac{1}{k}$, Error bars, reported only for weighting factor = 0.35 for readability
reasons, denote the error of the estimate calculated as standard deviation of the whole Markov chain and depends
on the model structure, model assumptions and priors.


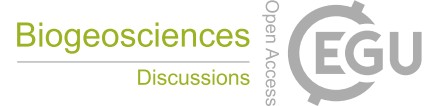

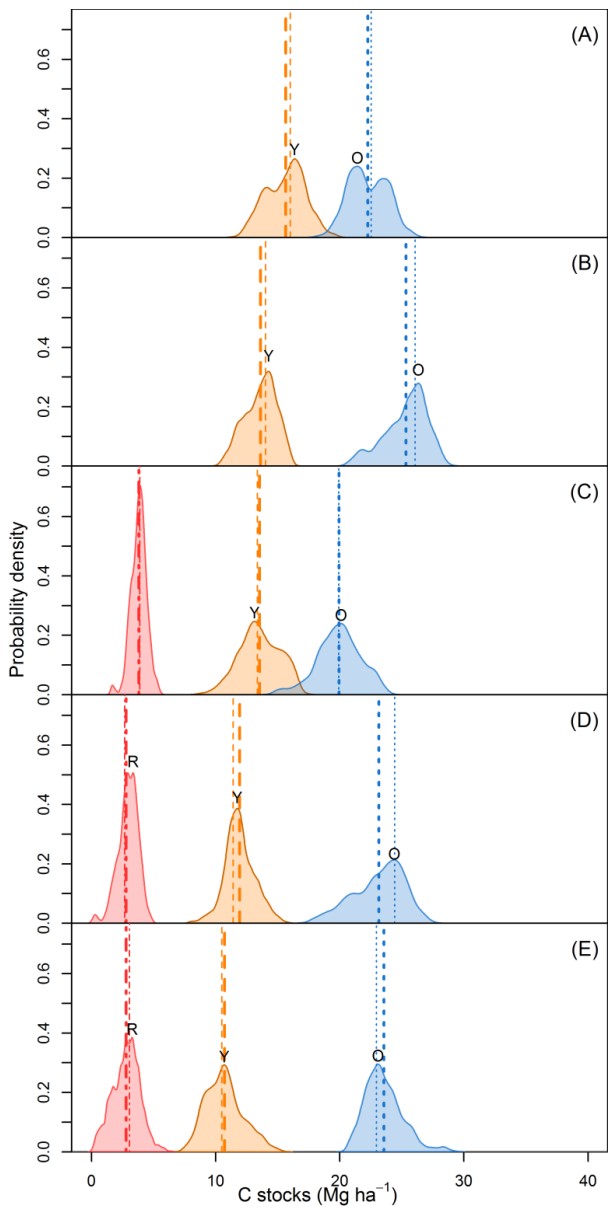

Figure 9: Probability distribution of the initial size of the C pools (Y=Young, O=Old, R=Recalcitrant) in
structure I (A), IV (B), II (C) and V (D), with weighting factor = 0.35. On the vertical axis is depicted the
probability density of the parameter (dimensionless) and on the horizontal axis the value of the parameter (in Mg
ha$^{-1}$). Vertical lines are representing the mean value (thick lines) and the Venter estimated mode (thin lines) of
the Markov chains.

