# Peer review of "Parametrization consequences of"

_Biogeosciences, 2015_

## Referee Comment (RC1) · Anonymous Referee #1 · 22 Feb 2016

This paper discuss the use of SOC and SO14C by changing relative weighing of these two data and by changing model structure. I enjoyed this paper a lot and I think this paper has value to be published in this journal. I agree that we have still lack data for sustaining more detailed mechanistic models. I have following technical comments.

Page 6, Line 14 (P6L14): 56 kg ha-1 -> -1 should be superscript. Same for line 15 and 16. P6L16: 1 t -> 1 Mg P6L18-19: Are these properties of topsoil? Is so, please mention it and please describe the depth of topsoil. P6L21: and by-products -> and above-ground part of by-products ? Is it OK? If so, please describe what happened with below-ground part. Was it incorporated into soil? or removed? P7L8: Levin, Ingeborg and Kromer 2004 -> Levin and Kromer 2004. Same for line 10-11 P10L15-16: Maize is

C4 plant. Is it OK to neglect this? P13L7: S3 appears here before S2 appears. Please arrange them in order as it appears in the text. P13L21-22: 10.000, 5000, 100.000 -> 10,000, 5,000, 100,000 ? Please unify notation. P16L9: Fig. 8 appears here before figs 3 to 7. Please arrange in a order. P17L4-18: I think this part should be moved to "Results" parts. The beginning of section "3.3" may be good. P19L24-25: ")" after "feedbacks" ? P22L14: "I, II and II" may be wrong. Please check and correct. P25L2: Bolinder, M. a. et al., -> Please put all authors names. Table1: Unit of SOC should be Mg ha-1, not Mg ha-1 y-1. Please put information of depth to which SOC values were calculated. "*" is not in this table though "*=average" indication. N2P2K2Mg -> Please use subscript for numbers, or indicate just "Chemical fertilizer". Figure3: Please explain what A - F indicate. Figure9: What is "E"?

---

## Referee Comment (RC2) · Anonymous Referee #2 · 6 Mar 2016

Review of "Parameterization consequences of constraining soil organic matter models by total carbon and radiocarbon using longterm field data" by Menichetti et al.

This study used field experiment SOC and SO14C data to test/constrain five different soil decomposition model structures that explicated represented total SOC and 14C pools. Their main conclusions include 1) according to model, the treatment sites SOC were far from equilibrium. 2) model estimated mean MRT was sensitive to the consideration of 14C data. 3) estimated parameters were less sensitive to model structures. 4) The data were not sufficient to determine an optimal mechanistic SOC model structure.

I appreciate the authors effort in utilizing 14C data and the uncertainty analysis using different model structures. I do advocate the additional constraints imposed by 14C,

which is crucial for getting the turnover time right.

General comments: The five models are difference models, I am wondering why do the authors not choose to solve a system of Ordinary Differential Equations, because difference equation systems can yield in notable differences from the ODEs.

The thickness of the soil that correspond to the SOC measurement was not mentioned in the text. How is the 14C measurement, presumably from a mixtures of soil samples of both shallow and deep soil, differentiated and associated with the modeling pools.

The details of the 'time series' of SOC and 14C data are not clear; it is also not clear what cost/target function were used in the optimization process. A graph showing the time evolving dynamics of the observations would help. Based on the text, it seems the measurement of SOC and 14C were not vertically resolved, then it is very critical to report the depth to which SOC was sampled. If only total SOC and carbon-averaged 14C is measured for one treatment, then the model seems to have too many parameters (minimum 4 parameters) than the observations (total SOC and 14C, two values).

The MRT of the old pool obtained from the model is <100 years, while many studies reporting deep soil turnover of thousands of years (Schelesinger 2000), thus knowing the depth of the sampling is important to understand the magnitude of turnover time in this crop ecosystem.

Specific comments: Abstract: the reported MRT have too many significant digits, unless a range is able to be reported, otherwise please consider round to the units. Actually if the authors were running 4 MC chains for each calibration, then one should be able to calculate a variance for the MRT.

Abstract: where is there no uncertainty range on the estimated MRT?

P4 L23: change to "are currently under active development"

The exact time span and frequency of the SOC and 14C measurements should be specified in section 2.2, currently it is not clear about this data information

P8 L15: the notation 'i' was not defined in the text, seems it is inputs to the young pool. Then this model assumes no direct inputs to the old pool, but there are root exudates etc. that maybe directly input to the old pool. The current model structure may be fine, but some discussion on this potential caveat may be helpful. This is important because the 14C from root exudates are usually very young, thus maybe have a big impact on the 14C signature of the old pool.

Fig 1: consider relocate the legend to not interfere with the lines.

Fig 2: It is surprising to see the graph that shows the impact of different weights of SOC vs. 14C on the model optimization results. It is interesting to see that model structure has such a limited effect on the results, suggesting current dynamics in the models are not being well differentiated at all. One potential reason is that the limited 14C and SOC observations (not depth resolved and not really time series? Please see the general comment) are not sufficient to distinguish model structure, I would expect one to see stronger differentiation when vertically resolved SOC and 14C information was incorporated.

Fig 7: just by reading the caption, it is not clear what the different panels represent. Please clarify. Now I see what the observation look like. The high 14C in the observation makes me wonder that the 14C observation may not be sufficient to constrain the turnover of the old soil carbon, as the result shows, that the different model structures are not distinguishable. Maybe it is worth mentioning in the discussion the importance of vertically resolved SOC and 14C observations in model parameterization.

---

## Short Comment (SC1) · 9 Mar 2016

We are grateful to the referee fort the comments and the improvement they will bring to the manuscript, and glad that the manuscript was appreciated.

**P6L14, P7L8, P19L24-25, P22L14, P25L2, Table 1.** All the comments concerning typographic issues, units and references will be dealt with according to the comments.

**P6L18-19.** The properties indicated are from the topsoil. The study concerns just the topsoil (0-25 cm), and this is so far not adequately specified in the paper as pointed out also by referee 2. We will provide to describe this in detail.

[Figure]

**P6L2.** Also aboveground byproducts are removed, while belowground production is incorporated into the soil. This will be described better.

**P10L15-16.** If the referee refers to the different kinetic fractionation of isotopes in maize compared to $C_3$ plant due to the malate-aspartate pathway, this should not influence the $^{14}C$ signature because of $^{13}C$ normalization. The $^{13}C$ normalization is a standard procedure in $^{14}C$ data reporting, and has been considered in the manuscript. The $^{13}C$ data are used to take care of any fractionation due to chemical kinetic effects and to filter these effects out from the $^{14}C$ signal. This is true also for the mechanism associated to the photosynthetic pathways.

**P13L7.** Order of supplements will be rearranged.

**P13L21-22.** Notations will be standardized.

**P17L4-18.** We will consider moving the indicated section to the results, and we judge this a really useful suggestion.

**Figure3.** The letters refer to the different parameters, as indicated also on the y axis. Letters to indicate subpanels have been utilized as best practice, although they are not utilized in the text. We believe it can be useful to have them there for future references. We will specify in the caption what the letters refer to.

**Figure9.** Caption is wrong, referring to a former version of the same figure. Sorry for the mistake. The panels are now referring to structure I (A), II (B), III (C), IV (D) and V (E). This will be modified.

---

## Short Comment (SC2) · 11 Mar 2016

We are really grateful for the suggestions, and for pointing out a few clear mistakes giving us the possibility to address them.

**1 Generic comments**

**1.1 Recursive equations**

Regarding the choice of using recursive equations rather than ODEs, it has been motivated mostly by convenience in the implementation. This choice allowed us to run a single parameter set in a way that was much faster than by utilizing at each run an ODE solver, and therefore helped greatly our study since it reduced the time for a single run of the calibration to few hours. The choice of running the equation in recursive steps helps also to simplify the use of the recent atmospheric $^{14}C$ profile since 1950 (which is highly nonlinear and requires the model to run in steps). One of the advantages of a model on the minimalistic side like ICBM is that there is an analytical solution, which has been given in the form of recurrence equation by Kätterer (2004). Since this solution is analytical and not an approximated numerical solution (and it is therefore independent from the parameter set), the results are consistent.

**1.2 Thickness of the soil layer considered**

The thickness of the soil considered is for sure a crucial parameter, and we forgot to describe this detail in the text. The depth considered was always 25 cm, since here we aimed at modelling the topsoil influenced by the cultivation practices. The mechanical ploughing in ZOFE is done down to that depth. This information will be introduced in the text.

**1.3  Discussing the MRT estimates**

The depth is probably one of the main reasons for the difference in the MRT estimate of the "old" pool as compared to other studies, since we are not considering deep layers where SOC is stabilized by many processes and thousands of years old. Eventually also the definition of the pools, which is dependent on the model structure chosen, should be considered as a possible concurrent explanation. But in this case we believe the main point to consider is the depth, as pointed out by the referee.

**1.4  Optimization**

The cost function utilized was the default likelihood function in JAGS and/or WinBUGS framework (it refers to the likelihood of the parameters given the observations and it is proportional to RMSE) as well as the default search algorithm (a basic Metropolis-Hastings search).

**1.5  Other comments**

The time series (observations) are shown entirely in Figures 6 and 7.

**2  Specific comments**

1. Thanks for the comment. This will be modified.

2. Thanks for the comment. We will introduce the uncertainty also in the abstract.

3. The time span and frequency of the measurements is irregular, as often the case in multi-decadal experiments. The time series are configured therefore as irregular time series, and are treated accordingly. It is partially described in the text and in the relative references, but it is shown graphically in detail in Figures 6 and 7, where each measurement point is represented. Text will be made clearer.

4. That is correct, "$i$" denotes the inputs to the $Y$ pool only. The idea of considering inputs directly in the $O$ ("old") pool is interesting, but it might stem from a different understanding of the pool definition from the one in this manuscript. Since "young" and "old" are in these kinds of models defined essentially by their MRT, all the material is supposed to go through some sort of "humification" before passing to the "old" pool. This is valid even more for fast cycling material like exudates, but it seems valid also for fine roots in pores, for example. In this particular conceptual model, if some input C material is young this very basic property inherent to the material (its age) configures it automatically as grouped into the $Y$ pool. The development of SOC models with more mechanistic definition of the pools would allow among other things also for the incorporation and test of such hypothesis, and such development is indeed a fascinating idea although outside the scope of this manuscript.

5. Figure will be modified (by increasing the limits on the y axis)

6. This is indeed a comment straight to the point. The authors agree, and expect exactly the same thing and will hopefully proceed with testing also this hypothesis if time, funding and other constrain will allow us. More specifically, though, adding more data is expected in any case to improve the resolving power of the model but one of the problem we would face is how to define vertical processes and to decide on their level of abstraction. The increased model complexity when adding one spatial dimension will drive the results in the opposite direction (reducing

the definition and increasing parameter uncertainty), and the final result will be determined both by the added complexity (causing less definition) and the added information (causing more definition). And the way we will represent the spatial processes will also influence the result. In general, though, we expect results in line with this statement.

7. Captions for Figure 6 and 7 will be made more explicit. The possibility of a vertically resolved model will be mentioned explicitly in the text (Discussion section) as a possible future development.

---

## Author Comment (AC1) · 30 Mar 2016

We are grateful to the referee fort the comments and the improvement they will bring to the manuscript, and glad that the manuscript was appreciated.

**P6L14, P7L8, P19L24-25, P22L14, P25L2, Table1** All the comments concerning typographic issues, units and references will be dealt with according to the comments and resolved.
**P6L18-19** The properties indicated are from the topsoil. The study concerns just the topsoil (0-25 cm), and this is so far poorly specified in the paper as pointed out also by

referee #2. We will provide a more detailed description upon revision.

**P6L2** Also aboveground by-products are removed, while below-ground production is incorporated into the soil. This will be described better.

**P10L15-16** If the referee refers to the different kinetic fractionation of isotopes in maize compared to C3 plants due to the malate-aspartate pathway, this should not influence the $^{14}C$ signature because of $^{13}C$ normalization. The $^{13}C$ normalization is a standard procedure in $^{14}C$ data reporting, and has been considered in the manuscript. The $^{13}C$ data are used to take care of any fractionation due to chemical kinetic effects and to filter these effects out from the $^{14}C$ signal. This is true also for the mechanism associated to the photosynthetic pathways.

**P13L7** Order of supplements will be rearranged.

**P13L21-22** Notations will be standardized.

**P17L4-18** We will consider moving the indicated section to the results, and we appreciate this really useful suggestion.

**Figure3** The letters refer to the different parameters, as indicated also on the y axis. Letters to indicate subpanels have been utilized as best practice, although they are not utilized in the text. We believe it can be useful to have them there for future references. We will specify in the caption what the letters refer to.

**Figure9** Caption is wrong, referring to a former version of the same figure. We apologize for the mistake. The panels are referring to structure I *(A)*, II *(B)*, III *(C)*, IV *(D)* and V *(E)*. This will be modified.

---

## Author Comment (AC2) · 30 Mar 2016

We are really grateful for the suggestions, and for pointing out a few clear mistakes giving us the possibility to address them.

**1 Generic comments**

**1.1 Recursive equations**

Regarding the choice of using recursive equations rather than ODEs, this simplifies the implementation of the recent atmospheric $^{14}C$ profile since 1950 (which is highly non-linear and requires the model to run in steps anyway) and the utilization of an irregular input time series. This choice allowed us to run a single parameter set in a way that was much faster than by utilizing at each run and ODE solver, therefore helping greatly our study since this reduced the time for a single run of the calibration to few hours. One of the advantages of a model on the minimalistic side like ICBM is that there is an analytical solution, which has been given in the form of recurrence equation by Kätterer (2004). Since this solution is analytical and not an approximated numerical solution (and it is therefore independent from the parameter set), the results are consistent.

**1.2 Thickness of the soil layer considered**

The thickness of the soil considered is for sure a crucial parameter, and we forgot to describe this detail in the text. The depth considered was always 25 cm, since here we aimed at modelling the topsoil influenced by the cultivation practices. The mechanical ploughing in ZOFE is done down to that depth. This detail will be added in the text.

**1.3 Discussing the MRT estimates**

The depth is probably one of the main reasons for the difference in the MRT estimate of the "old" pool as compared to other studies, since we are not considering deep

layers where SOC is stabilized by many processes and thousands of years old. Eventually also the definition of the pools, which is dependent on the model structure chosen, should be considered as a possible concurrent explanation. But in this case we believe the main point to consider is the depth, as pointed out by the referee.

**1.4 Optimization**

The cost function utilized was the default likelihood function in JAGS and/or WinBUGS framework (it refers to the likelihood of the parameters given the observations and it is Gaussian) as well as the default search algorithm (a basic Metropolis-Hastings search). This will be better specified in the text.

**1.5 Other comments**

The time series (observations) are shown entirely in Figures 6 and 7.

**2 Specific comments**

1. Thanks for the comment. This will be modified.

2. Thanks for the comment. We will introduce the uncertainty also in the abstract.

3. The time span and frequency of the measurements is irregular, as often the case in multi-decadal experiments. The time series are configured therefore as irregular time series, and are treated accordingly. It is partially described in the text and in the relative references, but it is shown graphically in detail in Figures 6 and 7, where each measurement point is represented. Text will be made clearer.

4. That is correct, "$i$" denotes the inputs to the $Y$ pool only. The idea of considering inputs directly in the $O$ pool is interesting, but it might stem from a different understanding of the pool definition from the one in this manuscript. Since $Y$ and $O$ are in these kinds of models defined essentially by their MRT, all the material is supposed to go through some sort of "humification" before passing to the $O$ pool. This is valid even more for fast cycling material like exudates, but it seems valid also for fine roots in pores, for example. In this particular conceptual model, if some input C material is young this very basic property inherent to the material (its age) configures it automatically as grouped into the $Y$ pool. The development of SOC models with more mechanistic definition of the pools would allow among other things also for the incorporation and test of such hypothesis, and such development is indeed a fascinating idea although outside the scope of this manuscript.

5. Figure will be modified (by increasing the limits on the y axis)

6. This is indeed a comment straight to the point. The authors agree, and expect exactly the same thing and will proceed with testing also this hypothesis in the future. More specifically, though, adding more data is expected in any case to improve the resolution of the model, but one of the problem we would face is how to define vertical processes and to decide on their level of abstraction. The increased model complexity when adding one spatial dimension will drive the results in the opposite direction (reducing the definition and increasing parameter uncertainty), and the final result will be determine both by the added complexity (causing less definition) and the added information (causing more definition). And the way we will represent the spatial processes will also influence the result. In

general, though, we expect results in line with this statement.

7. Captions for Figure 6 and 7 will be made more explicit. The possibility of a vertically resolved model will be mentioned explicitly in the text (Discussion section) as a possible future development.

---

## Author Response (AR1)

**Answers to the referees:**

**Referee #1**

**P6L14, P7L8, P19L24-25, P22L14, P25L2, Table1** All the comments concerning typographic issues, units and references have been dealt with.

**P6L18-19** The properties indicated are from the topsoil. The study concerns just the topsoil (0-25 cm), and this was so far poorly specified in the paper as pointed out also by referee #2. This is now described in the text.

**P6L2** Aboveground byproducts are removed. Belowground byproducts are represented only by roots when the rhizome is harvested (e.g. potatoes or beetroots), and are incorporated back into the soil. This is now described in the text.

**P7L8** This has been corrected.

**P10L15-16** If the referee refers to the different kinetic fractionation of isotopes in maize compared to $C_3$ plants due to the malate-aspartate pathway this should not influence the $^{14}C$ signature because of $^{13}C$ normalization. The $^{13}C$ normalization is a standard procedure in $^{14}C$ data reporting, and has been considered in the manuscript. The $^{13}C$ data are used to take care of any fractionation due to chemical kinetic effects and to filter these effects out from the $^{14}C$ signal. This is true also for the mechanism associated to the photosynthetic pathways.

**P13L7** Order of supplements has been rearranged.

**P13L21-22** Notations have been standardized.

**P17L4-18** The indicated section has been moved as suggested.

**Figure3** The letters refer to the different parameters, as indicated also on the y axis. Letters to indicate subpanels have been utilized as best practice, although they are not utilized in the text. We believe them to be useful for future references. This is now specified in the caption.

**Figure9** Caption is wrong, referring to a former version of the same figure. We apologize for the mistake. The panels are now referring to structure I (A), II (B), III (C), IV (D) and V (E). This has been modified.

**Referee#2**

Regarding the choice of using recursive equations rather than ODEs, this is motivated by convenience in the implementation. This choice allowed us to run a single parameter set in a way that was much faster than by utilizing at each run an ODE solver, therefore helping greatly our study since this reduced the time for a single run of the calibration to few hours. The choice of running the equation in recursive steps helps also to simplify the implementation of the recent atmospheric $^{14}$C profile since 1950 (which is highly nonlinear and requires the model to run in steps anyway). One of the advantages of a model on the minimalistic side like ICBM is that there is an analytical solution, which has been given in the form of recurrence equation by Kätterer (2004). Since this solution is analytical and not an approximated numerical solution (and it is therefore independent from the parameter set), the results are consistent.

The thickness of the soil considered is for sure a crucial parameter, and we forgot to describe this detail in the text. The depth considered was always 25 cm, since here we aimed at modelling the topsoil influenced by the cultivation practices. The mechanical ploughing in ZOFE is done down to that depth. This detail is now in the text.

The depth is probably one of the main reasons for the difference in the MRT estimate of the "old" pool as compared to other studies, since we are not considering deep layers where SOC is stabilized by many processes and thousands of years old. Eventually also the definition of the pools, which is dependent on the model structure chosen, should be considered as a possible concurrent explanation. But in this case we believe the main point to consider is the depth, as pointed out by the referee, and it is now understandable by the reader.

The cost function utilized was the default likelihood function in JAGS and/or WinBUGS framework (it refers to the likelihood of the parameters given the observations and it is Gaussian) as well as the default search algorithm (a basic Metropolis-Hastings search). This is not better specified in the text.

The time series (observations) are shown entirely in Figures 6 and 7.

Specific comments:

and 2) Thanks for the comments. This has been modified, and uncertainty is now reported also in the abstract.

3) The time span and frequency of the measurements is irregular, as often the case in multi-decadal experiments. The time series are configured therefore as irregular time series, and are treated accordingly. It is partially described in the text and in the relative references, but it is shown graphically in detail in Figures 6 and 7, where each measurement point is represented. The irregularity of time series is now explicit in the text.

4) That is correct, "i" denotes the inputs to the "young" pool only, and this is now explained in the text.

The idea of considering inputs directly in the "old" pool is interesting, but it might stem from a different understanding of the pool definition from the one in this manuscript. Since "young" and "old" are in these kinds of models defined essentially by their MRT, all the material is supposed to go through some sort of "humification" before passing to the "old" pool. This is valid even more for fast cycling material like exudates, but it seems valid also for fine roots in pores, for example. In this particular conceptual model, if some input C material is young this very basic property inherent to the material (its age) configures it automatically as grouped into the "young" pool. The development of SOC models with more mechanistic definition of the pools would allow among other things also for the incorporation and test of such hypothesis, and such development is indeed a fascinating idea although outside the scope of this manuscript.

5) Figure has been modified (by increasing the limits on the y axis)

6) This is indeed a comment straight to the point. The authors agree, and expect exactly the same thing and will proceed with testing also this hypothesis in the future. More specifically, though, adding more data is expected in any case to improve the resolution of the model, but one of the problems we would face is how to define vertical processes and to decide on their level of abstraction. The increased model complexity when adding one spatial dimension will drive the results in the opposite direction (reducing the definition and increasing parameter uncertainty), and the final result will be determine both by the added complexity (causing less definition) and the added information (causing more definition). And the way we will represent the spatial processes will also influence the result. In general, though, we expect results in line with this statement. This view is now represented in the text, at the end of the discussion.

7) Captions for Figure 6 and 7 have been made more explicit. The possibility of a vertically resolved model is now mentioned explicitly in the text (discussion section) as a possible future development.

[revised manuscript text omitted]